



Geoscientific
Model Development

# A new open-source viscoelastic solid earth deformation module implemented in Elmer (v8.4)

**Thomas Zwinger**[1], **Grace A. Nield**[2,3], **Juha Ruokolainen**[1], **and Matt A. King**[2]

[1]CSC–IT Center for Science Ltd., Espoo, Finland
[2]Surveying and Spatial Sciences, School of Technology, Environments and Design, University of Tasmania, Hobart, Australia
[3]Department of Geography, Durham University, Durham, UK

**Correspondence:** Thomas Zwinger (zwinger@csc.fi)

**Abstract.** We present a new, open-source viscoelastic solid earth deformation model, Elmer/Earth. Using the multi-physics finite-element package Elmer, a model to compute viscoelastic material deformation has been implemented into the existing linear elasticity solver routine. Unlike approaches often implemented in engineering codes, our solver accounts for the restoring force of buoyancy within a system of layers with depth-varying density. It does this by directly integrating the solution of the system rather than by applying stress-jump conditions in the form of Winkler foundations on inter-layer boundaries, as is usually needed when solving the minimization problem given by the stress divergence in commercial codes. We benchmarked the new model with results from a commercial finite-element engineering package (ABAQUS, v2018) and another open-source code that uses viscoelastic normal mode theory, TABOO, using a flat-earth setup loaded by a cylindrical disc of 100 km in CE1 diameter and 100 m in height at the density of ice. Evaluating the differences in predicted surface deformation at the centre of the load and two distinctive distances (100 and 200 km), average deviations of 7 and 2.7 cm of Elmer/Earth results to ABAQUS and TABOO, respectively, were observed. In view of more than 100 cm maximum vertical deformation and the different numerical methods and parameters, these are very encouraging results. Elmer is set up as a highly scalable parallel code and distributed under the (L)GPL license, meaning that large-scale computations can be made without any licensing restrictions. Scaling figures presented in this paper show good parallel performance of the new model. Additionally, the high-fidelity ice-sheet code Elmer/Ice utilizes the same source base as Elmer and thereby the new model opens

the way to undertaking high-resolution coupled ice-flow–solid-earth deformation simulations, which are required for robust projections of future sea-level rise and glacial isostatic adjustment.

## 1 Introduction

Reconstructing ice-sheet history and predicting ice-sheet response to changes in climate are imperative for accurately predicting future ice-mass loss and hence sea-level rise. An important component of ice-sheet evolution is the isostatic response of the solid earth that occurs as a result of changes in the mass of the ice sheet. Over glacial cycles the waxing and waning of ice sheets causes the underlying earth to deform as the ice loading at the surface grows and shrinks. This deformation occurs both instantaneously as an elastic response and over longer timescales as the viscous mantle flows back to previously glaciated regions in order to regain gravitational equilibrium. How fast or slowly the earth deforms depends on the underlying mantle viscosity, and, although typically thought to occur over several thousands of years (Whitehouse, 2018, and references therein), recent studies have shown regions undergoing much more rapid (decadal) rebound in response to present-day changes (Nield et al., 2014; Barletta et al., 2018).

This isostatic response of the bedrock can strongly influence ice-sheet dynamics. Deformation of the earth changes the elevation of the ice sheet which in turn affects the surface temperature and the rate of accumulation or ablation. Solid earth deformation also alters the gradient of the bedrock on

which the ice sheet rests, particularly at the periphery, altering the internal forces as well as the driving stress and therefore the flow of the ice sheet (Le Meur and Huybrechts, 1996; Adhikari et al., 2014). In marine-grounded ice sheets lying on a reverse slope bed (e.g. West Antarctica) these effects can be critical. As the grounding line retreats further along the reverse slope into deeper water, ice flux across the grounding line increases leading to increased loss (Schoof, 2007). However, bedrock uplift can have a stabilizing effect by reducing the slope of the reverse bed and thereby slowing the retreat of the grounding line (Gomez et al., 2010, 2013).

Including the isostatic response of bedrock in an ice-sheet model is therefore crucial to obtaining accurate predictions of ice-sheet mass balance, and there are several methods which can be used. Computing the isostatic response with a self-gravitating viscoelastic spherical earth is the most accurate, but most computationally expensive, method. Several simple approximations are often made using models with a combinations of local lithosphere or elastic lithosphere with diffusive asthenosphere or relaxing asthenosphere (Le Meur and Huybrechts, 1996; Rutt et al., 2009). Of these, Le Meur and Huybrechts (1996) found the best performing is the "ELRA" (e.g. Greve, 2001) model (elastic lithosphere with relaxing asthenosphere) which is widely used in ice-sheet modelling, mainly due to its simplicity and fast computations. However, Bueler et al. (2007) found significant differences in resulting bed elevation and ice-sheet thickness when using a model with ELRA compared to a spherical self-gravitating model due to the shortcomings of using a constant relaxation time for the mantle as opposed to mode-dependent relaxation times (Peltier, 1974).

A further improvement to an ice-sheet model can be made by coupling a model of solid earth deformation to the ice-sheet model. Studies have demonstrated that the feedback between the two systems can have large impacts on ice-sheet evolution (Gomez et al., 2013; de Boer et al., 2014). Using a coupled model Gomez et al. (2015) showed a reduced estimate of Antarctic ice-mass loss compared with a model without solid earth effects included. However, due to the large computational expense of these models, they remain at a relatively low resolution both spatially and temporally therefore omitting short wavelength and short timescale deformations. A recent study by Larour et al. (2019) showed that models need kilometre-scale resolution in the horizontal components to accurately predict ice-sheet evolution in the region of ice-sheet mass change, particularly for the short wavelength elastic component of solid earth deformation. This demonstrates the clear need for a full Stokes ice-sheet model capable of computing high resolution solid earth rebound.

Wu (2004) presented a recipe to adapt existing commercial finite-element codes to compute earth deformation as a response to ice loads, both for flat-earth as well as spherical self-gravitating setups. Finite elements have the advantage that they in general can use unstructured meshes in order to provide the resolution needed in regions where either physics or geometry demand it while keeping the model size limited. Many finite-element packages also include versatile solution methods that often also work in parallel computing environments – an essential feature to address continental-size problems at high resolution.

## 2 Mathematical and numerical model

The implementation of the viscoelastic rheology and additional force terms, to a large extent, follows the one suggested by Wu (2004). Adopting their notation, we start from the viscoelastic stress tensor, $\boldsymbol{\tau}$ defined by the differential equation

$$\frac{\partial \boldsymbol{\tau}}{\partial t} = \frac{\partial \boldsymbol{\tau}_0}{\partial t} + \frac{\mu}{\nu}\left(\boldsymbol{\tau} - \Pi\mathbf{1}\right), \tag{1}$$

with the stress $\boldsymbol{\tau}_0$ in the case of incompressibility given by

$$\boldsymbol{\tau}_0 = \Pi\mathbf{1} + 2\mu\boldsymbol{\epsilon}, \tag{2}$$

where $\Pi$ denotes the isotropic part of the Cauchy stress, i.e., the pressure. In the derivatives of Eqs. (1) and (2), $t$ stands for time, $\mathbf{1}$ denotes the unit-tensor, $\mu$ the shear modulus and $\nu$ is the viscosity. The strain-tensor $\boldsymbol{\epsilon}$ written in terms of the displacement $\boldsymbol{d}$ denotes as

$$\boldsymbol{\epsilon} = \mathrm{sym}(\nabla\boldsymbol{d}) = \frac{1}{2}\left(\nabla\boldsymbol{d} + (\nabla\boldsymbol{d})^{\mathrm{T}}\right). \tag{3}$$

The linearized equation of motion for solid earth deformation (Wu, 2004) is given by

$$\nabla \cdot \boldsymbol{\tau} - \nabla(\rho_0 \boldsymbol{g}_0 \dot{d}) - \rho_1 \boldsymbol{g}_0 - \boldsymbol{g}_0 \nabla\phi_1 = \mathbf{0}. \tag{4}$$

Where $\rho_0$ and $\boldsymbol{g}_0$ are hydrostatic background density and gravity, respectively, and $\rho_1$ is the perturbed density. The direction of $\boldsymbol{g}_0$ is in negative radial direction. According to Wu (2004, Sect. 3) a flat-earth model is derived from Eq. 4 [TS1] by assuming incompressibility and ignoring self-gravitational effects (i.e., redistribution of mass), making the third and fourth terms vanish. Further, sphericity is ignored, leading to changes aligned with the unit vector of a Cartesian system in vertical direction, $\boldsymbol{e}_z$. This leads to the equation of motion for a non-self-gravitating flat-earth model with layer-wise constant material. It reduces to a balance between the divergence of the stress (first term) and a restoring force due to the advection of pre-stress of the material (Wu, 2004)

$$\nabla \cdot \boldsymbol{\tau} - \rho g \nabla(\boldsymbol{e}_z \cdot \boldsymbol{d}) = \mathbf{0}. \tag{5}$$

Here, $\rho = \rho_0$ and $g = ||\boldsymbol{g}_0||$ is the magnitude of the local acceleration by gravity, which points into the negative direction of $\boldsymbol{e}_z$.

## 2.1 Implementation in Elmer/Earth

Elmer/Earth is based on the open-source finite-element package Elmer (Råback et al., 2019). In order to build a flat-earth model as described in the previous section, Eq. (1) has been added to the existing linear elasticity solver of Elmer. In the case of incompressibility, the additional variable of pressure, $\Pi$, has been introduced to the solver. This avoids the singularity of the compressible formulation in the case of the Poisson ratio approaching 0.5.

Many commercial codes lack an implementation of the second term in Eq. (5), which implies a transformation of the stress to reduce the formulation to only the first term. As a consequence of this stress transformation, additional jump conditions in the form of Winkler foundations (Wu, 2004) have to be imposed on internal boundaries that mark a jump in either the gravity or the density. This can be inconvenient in building the model, as a detailed description of the setup may contain boundaries for more than 10 layers.

Here we take advantage of the accessibility of the source code of Elmer by including this term in the weak formulation that uses the viscoelastic stress. The second term in Eq. (5) thereby contributes to the stiffness matrix. Naturally, the formulation still needs a layered structure of the model, i.e., material parameters are kept constant for certain layers. This can be easily achieved as Elmer allows material parameters to be prescribed as well as body forces (in our case gravity), on the basis of elements or even integration points (in addition to nodal values). This means that we are able to impose discontinuities in parameters over elements anywhere in the discretized computing domain without placing Winkler foundation boundaries at layer interfaces. In other words, no boundary conditions have to be set at internal layer boundaries. By including this term in the weak formulation of the problem, the method then automatically applies the restoring needed CE2 force on element boundaries with jumps in material properties or gravity, without the need to place boundaries in the mesh.

Discretization of the time derivatives for stress and pressure (in the case of incompressible material) is implemented by the first-order implicit difference

$$\frac{\partial \boldsymbol{\tau}}{\partial t} \approx \frac{\boldsymbol{\tau}^{i+1} - \boldsymbol{\tau}^{i}}{\Delta t}, \quad \frac{\partial \Pi}{\partial t} \approx \frac{\Pi^{i+1} - \Pi^{i}}{\Delta t}. \tag{6}$$

Here, $i$ is the current, and $i+1$ the implicit time step as well as $\Delta t = t^{i+1} - t^{i}$ the time-step size between. The solution of the time-evolution problem then reads

$$-\mathbf{1}\Pi^{i+1} + 2\mu\Phi\boldsymbol{\epsilon}^{i+1} = -\Phi\Pi^{i} + 2\mu\Phi\boldsymbol{\epsilon}^{i} - \Phi\boldsymbol{\tau}^{i}, \tag{7}$$

with $\phi = 1/(1+(\mu/\nu)\Delta t)$. The balance Eq. (5) of linear momentum is then solved for the new time step:

$$\nabla \cdot \boldsymbol{\tau}^{i+1}(\boldsymbol{d}) - \rho g \nabla \left(\boldsymbol{e}_z \cdot \boldsymbol{d}^{i+1}\right) = \mathbf{0}. \tag{8}$$

The weak formulation then results from the integral over the whole domain $\Omega$ (with its confining surface $\partial\Omega$) using the test and weighting function vectors $\boldsymbol{u}, \boldsymbol{v} \in H^1$:

$$\int_\Omega \boldsymbol{\tau}(\boldsymbol{u}) \cdot (\nabla \boldsymbol{v}) \, \mathrm{d}V - \oint_{\partial\Omega} (\boldsymbol{\tau}(\boldsymbol{u}) \cdot \boldsymbol{n}) \cdot \boldsymbol{v} \, \mathrm{d}A$$
$$- \int_\Omega \rho g \nabla (\boldsymbol{e}_z \cdot \boldsymbol{u}) \cdot \boldsymbol{v} \, \mathrm{d}V = 0. \tag{9}$$

Note that the divergence of the stress tensor has been partially integrated, leading – according to Green's theorem – to a term that integrates the stress vector, $\boldsymbol{t} = \boldsymbol{\tau}(\boldsymbol{u}) \cdot \boldsymbol{n}$, over $\partial\Omega$ with its surface normal $\boldsymbol{n}$. Taking additionally into account that $\boldsymbol{\tau}(\boldsymbol{u})$ is a symmetric tensor, only the symmetric part of $\mathrm{sym}(\nabla \boldsymbol{v}) = \boldsymbol{\epsilon}(\boldsymbol{v})$ contributes to the first integral, leading to the symmetric stiffness matrix in the weak formulation

$$\int_\Omega \boldsymbol{\tau}(\boldsymbol{u}) \cdot \boldsymbol{\epsilon}(\boldsymbol{v}) \, \mathrm{d}V - \oint_{\partial\Omega} (\boldsymbol{\tau}(\boldsymbol{u}) \cdot \boldsymbol{n}) \cdot \boldsymbol{v} \, \mathrm{d}A$$
$$- \int_\Omega \rho g \nabla (\boldsymbol{e}_z \cdot \boldsymbol{u}) \cdot \boldsymbol{v} \, \mathrm{d}V = 0. \tag{10}$$

The system is completed by boundary conditions that are either provided by a value for any component of the stress vector, $\boldsymbol{t} = \boldsymbol{\tau} \cdot \boldsymbol{n}$, in the second integral (Neumann condition) of Eq. (10) or by imposing a value for any component of the deformation vector, $\boldsymbol{d}$ (Dirichlet condition).

Equation (10) is solved using the standard Galerkin method with first-order basis functions in the case of the benchmark described in Sect. 3. Apart from this particular choice, Elmer provides a variety of possible basis functions left to the choice of the user. The iteration for the viscous contribution is computed on the Gaussian integration points. In the case of incompressibility, stabilization has to be applied by the residual free bubble method.

## 3 Benchmark tests

Benchmark tests are performed in order to validate the new implementation of Elmer/Earth in comparison to two other codes: ABAQUS and TABOO. We force the models with changing surface load, representing an idealized ice loading experiment. Specific geometry, earth structure and ice loading for the benchmarking case are described in Sect. 3.3. The two other codes are briefly introduced in the following sections.

### 3.1 Reference model ABAQUS

We use the finite-element software package ABAQUS (Hibbitt et al., 2016; software version 2018) to construct a model to verify the results of the new viscoelastic solver implemented in Elmer. We choose this approach to replicate the

Please note the remarks at the end of the manuscript.

geometry and equations implemented in the Elmer/Earth model as fully as possible. The model is a 3-D flat-earth model which computes the solid earth deformation in response to a changing surface load using the approach of Wu (2004). Buoyancy forces are accounted for by applying Winkler foundations to layer boundaries within the model where a density contrast occurs between two layers, and at the surface (Wu, 2004). The model has a large lateral extent to prevent boundary effects in the area of interest (Steffen et al., 2006) and has zero displacement imposed on its lateral and bottom boundaries. The model includes layers from the surface of the earth to the core–mantle boundary with parameters shown in Table 1.

## 3.2   Reference model TABOO

TABOO is an open-source post-glacial rebound calculator (Spada et al., 2003; Spada, 2003) that computes the deformation of the earth in response to a changing surface (glacial) load. The TABOO model assumes a spherically symmetric, incompressible earth with a Maxwell viscoelastic rheology (non-rotating, self-gravitational). TABOO implements the classical viscoelastic normal mode method commonly used in studies of glacial isostatic adjustment (Peltier, 1974). There are several inbuilt solid earth models available in TABOO with a specific earth structure and parameters and we use one of these for our synthetic benchmarking case study (Table 1, Sect. 3.3). Deformation is computed up to a user-specified spherical harmonic degree, and we chose 2048 (equivalent to approximately 10 km).

## 3.3   Test model setup

In order to test and compare the newly built Elmer/Earth model, a simple benchmark case has been set up for each of the models presented in Sect. 3.1 and 3.2. The benchmark case consists of a simple one-dimensional earth structure with parameters varying in the radial direction only, loaded and unloaded with a disc of ice. The models in Elmer/Earth and ABAQUS both use a flat-earth approximation, whereas TABOO is a fully spherical model. The effects of sphericity are negligible for the size of load we use for our benchmarking case. None of the models solve the "sea-level equation" (Farrell and Clark, 1976).

For the flat-earth approximation, the three-dimensional model domain stretches 4000 km in each horizontal direction from the centre of the ice load. This distance is 80 times the diameter of the test load, which is more than sufficient to allow mantle deformation below the ice load (Steffen et al., 2006). With depth, the model extends from the earth's surface at a radius of 6371 km to the core–mantle boundary with a total depth of 2891 km.

Geometry construction and meshing for Elmer/Earth simulations was achieved using the open-source software Gmsh (Geuzaine and Remacle, 2009). The lateral mesh resolution

for the ABAQUS model is a constant 10 km, whereas it varies for Elmer/Earth from 10 km for the area over which the load is applied to 200 km, increasing linearly, at the lateral domain boundaries (see Fig. 1a). The vertical resolution increases with depth as shown in Fig. 1b. The TABOO model has a spectral resolution equivalent to 10 km.

The earth structure used for the benchmarking case is one that is included as part of the TABOO package and is summarized in Table 1. The solid earth model consists of an elastic lithosphere, a viscoelastic upper mantle divided into three layers, and a viscoelastic lower mantle. Elmer/Earth applies incompressibility throughout the whole column and an extremely high viscosity of $\nu = 1 \times 10^{44}$ Pa s in the lithosphere, thereby enforcing an approximately elastic behaviour on the timescale of the load. This can be justified by the Maxwell time $t_\mathrm{M} = \nu/\mu$ being of the order of $10^{33}$ s, which indicates that viscous effects would only be significant at timescales several order of magnitudes larger than the timing of the load signal.

The viscosity of the upper and lower mantle is set to $1 \times 10^{18}$ and $1 \times 10^{22}$ Pa s, respectively, and the elastic and density parameters are depth-averaged values from the Preliminary Reference Earth Model (Dziewonski and Anderson, 1981; PREM). These parameters can easily be assigned to layers in both ABAQUS and Elmer. The relatively low value for the upper mantle helps to shorten the timescales for the benchmark test.

For the benchmark case we compute the deformation caused by an instantaneously imposed ice load at $t = 0$. Starting from an equilibrium bedrock with zero deformation, an ice load is instantaneously applied at the centre of the domain at the very beginning of the simulation. It is a 100 km diameter disc of 100 m height with a prescribed constant density of 917 kg m$^{-3}$. The load is maintained for 100 years after which it is instantaneously removed and the rebound computed for a further 100 years. The result on the vertical plane of symmetry from the reference run described in Sect. 5 is shown in Fig. 2.

The temporal evolution of the vertical displacement of the reference Elmer/Earth run (mesh1) over a line at the surface from the centre to the margin (0–200 km) is depicted in Fig. 3.

## 3.4   Numerical settings in Elmer/Earth

For all runs of Elmer/Earth presented in Sects. 4 and 5, the same numerical methods and parameters have been applied. A time-step size for the implicit backward differentiation formula (BDF) of the equivalent of 1 year has been chosen – in Sect. 5 we discuss the impact in accuracy by halving this time-step size. The resulting system matrix of the linear elasticity solver was first pre-conditioned using an ILU (incomplete lower–upper) factorization of first-order degree (ILU1, in Elmer terminology). To obtain a solution, its inverse was approximated using the GCR (generalized conju-

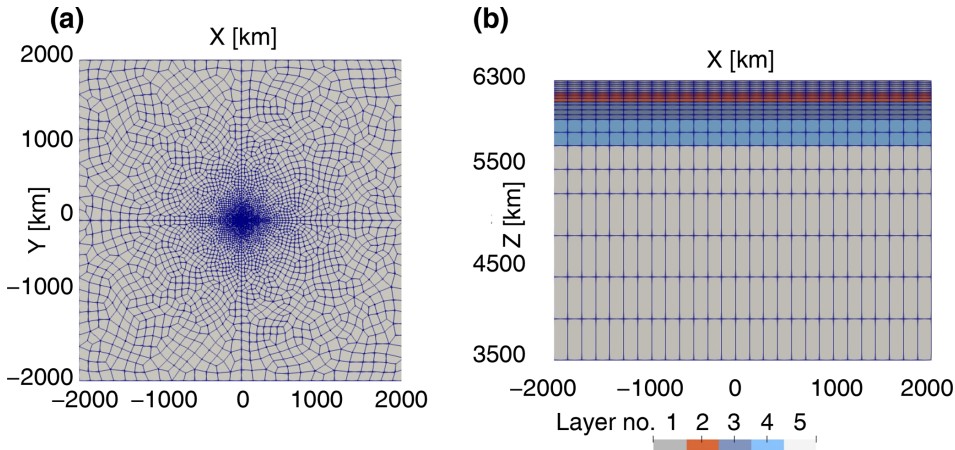

**Figure 1.** Top and side view of the reference run Elmer/Earth mesh (`mesh1`). The different layers corresponding to varying material parameters shown in panel **(b)** are given in Table 1. Annotated coordinates are in kilometres.

**Table 1.** Properties of the different layers in the flat-earth model benchmark. Vertical distances are with respect to earth's centre. The ABAQUS reference model uses a material model with a constant Poisson ratio of 0.49 throughout the whole domain.

| Layer | Vertical range (km) | Thickness (km) | $\varrho$ (kg m$^{-3}$) | $g$ (m s$^{-2}$) | $\rho$ (Pa s) | $E$ (Pa) |
|---|---|---|---|---|---|---|
| Lithosphere | 6371–6251 | 120 | 3233 | 9.87852 | 0 or $1 \times 10^{44}$ | $1.8388 \times 10^{11}$ |
| Upper mantle | 6251–6151 | 100 | 3367.12 | 9.939356456 TS2 | $\times 10^{18}$ | $1.9941 \times 10^{11}$ |
| | 6151–5971 | 180 | 3475.58 | 9.875562964 | $1 \times 10^{18}$ | $2.2948 \times 10^{11}$ |
| | 5971–5701 | 270 | 3857.75 | 9.839990347 | $1 \times 10^{18}$ | $3.1943 \times 10^{11}$ |
| Lower mantle | 5701–3480 | 2221 | 4877.91 | 9.792107051 | $1 \times 10^{22}$ | $6.5844 \times 10^{11}$ |

gate residual) Krylov subspace method (see, e.g., Eisenstat et al., 1983). A convergence criterion was applied for the relative norm of the solution vector between two iteration steps of $\varepsilon_d = 1 \times 10^{-7}$.

## 4 Comparison of results

Comparing the results of the benchmarking exercise with two models that use different methods gives us confidence in the implementation of the new Elmer code. Figure 4 shows displacement with time at three locations – the centre of the disc (indicated by 0 km) and at 100 and 200 km distance from the centre of the disc.

The displacement curves for all three models over major parts of the simulation agree to within an order of 10 cm (see Fig. 5) in relation to a maximum deformation of 1.1 m by ABAQUS at the centre. The largest difference is observed at the centre of the disc where the Elmer/Earth model deforms slightly less than ABAQUS and almost insignificantly more as TABOO but reaches this deformation more quickly than the other codes (i.e. it has a faster relaxation time). As a consequence, Fig. 5 shows differences in vertical displacement between models (also between ABAQUS and TABOO) to be

largest in the very beginning (when applying the load) and around the time of sudden unloading.

The small differences between the results could be caused by several factors. Mesh differences between Elmer/Earth and ABAQUS are the likely cause of some small differences with ABAQUS having a regular grid mesh and Elmer having a finer mesh at the centre of the disc. There seems to be a correlation of the resolution in the centre with the displacement in both FEM-based models. It seems that the ABAQUS model setup does not provide enough horizontal mesh resolution at the centre, where the load is applied. This is confirmed by results obtained with `mesh 2` (half mesh size) in Elmer/Earth, which produced displacements even larger than the one with the constant 25 km TS3 from ABAQUS (see Sect. 5).

The deformation calculated by TABOO is less than Elmer/Earth and ABAQUS at each location. This may be due to the fundamental differences in the computation methods employed by the TABOO code, implementing normal mode methods rather than finite-element methods. Furthermore, TABOO computes deformation on a self-gravitating solid earth, whereas ABAQUS and Elmer do not include self-gravitation, which would result in some differences between

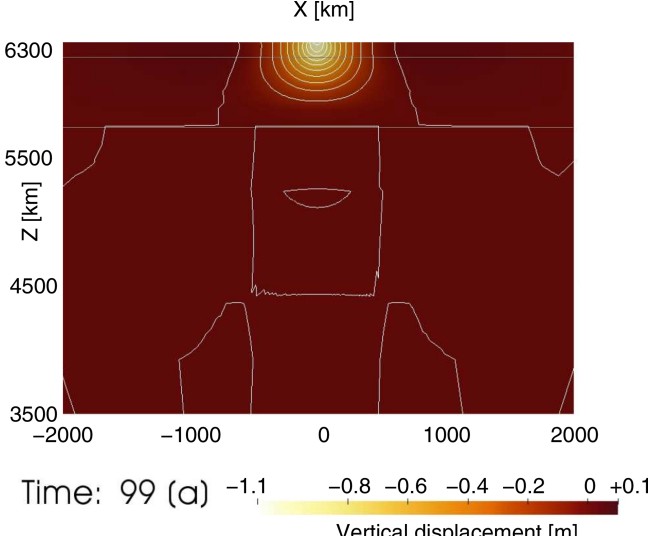

**Figure 2.** Cross section of the reference run with Elmer/Earth (`mesh1`) showing the vertical deformation at 99 years into the simulation at maximum deformation. Deformation is shown as colour texture as well as isoline (white in 0.1 m spacing). The boundaries between the lithosphere and upper and lower mantle (as given in Table 1) are annotated as grey lines. Annotated coordinates are in kilometres.

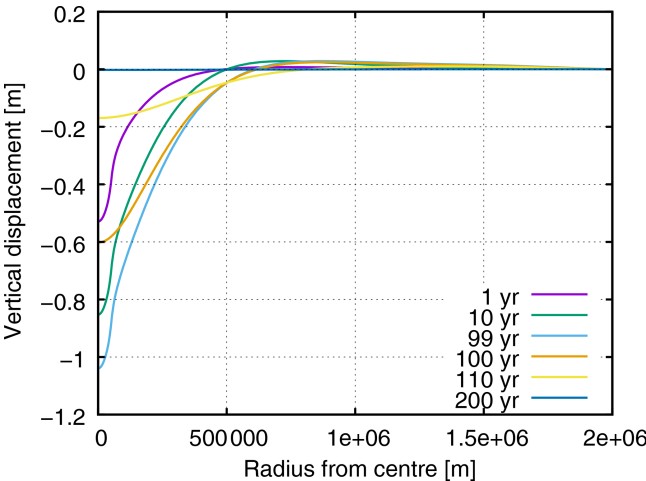

**Figure 3.** Temporal evolution of vertical displacement of the reference Elmer/Earth run (`mesh1`) over a line at the surface from the centre to the margin.

## 5 Performance and accuracy of the Elmer/Earth deformation model

In order to obtain some insight into parallel performance as well as the dependency on the mesh resolution of

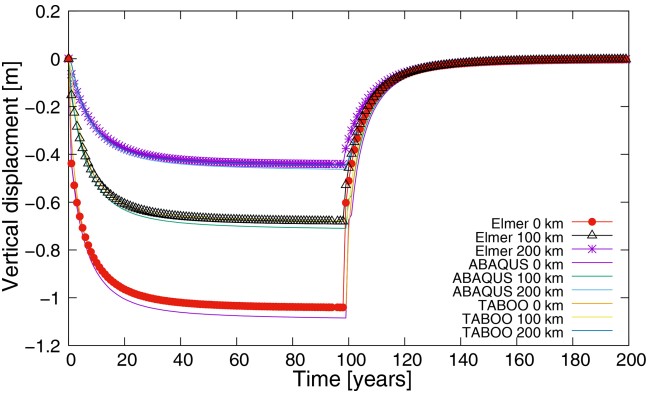

**Figure 4.** Comparison of results for deformation at the load centre (0 km), 100 and 200 km for Elmer/Earth, ABAQUS and TABOO.

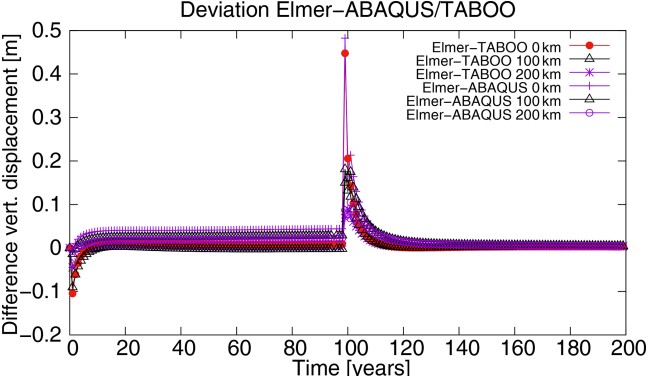

**Figure 5.** Difference in deformation of Elmer/Earth relative to ABAQUS and TABOO at the load centre (0 km), 100 and 200 km.

Elmer/Earth, three meshes with different resolutions and mesh partitions (4, 16 and 32) have been created (see Table 2). Partitioning of the meshes has been performed by the mesh-conversion program `ElmerGrid` (part of the Elmer installation) using the METIS k-way partitioning scheme (Karypis and Kumar, 1998).

Identical numerical parameters and methods, as described in Sect. 2, were applied throughout all runs.

### 5.1 Strong and weak scaling

Tests were performed on the Linux cluster `raijin` (Australian National Computational Infrastructure, 2017), utilizing compute nodes, each equipped with two Intel Xeon Sandy Bridge (E5-2670, 2.6 GHz) processors summing up to 16 cores per compute node. The code was compiled using the Intel compiler suite (version 2019.2.187) with Open MP (OMP) enabled, mainly to activate utilization of OMP-SIMD instructions within the code (Byckling et al., 2017). CPU-specific optimization was enabled by compiler flags `-O2 -march=sandybridge`. Basic linear algebra libraries (Lapack, BLAS, ScaLapack) were linked in from

**Table 2.** Parameters of the meshes and their partitions used for Elmer/Earth test runs.

| Mesh name | No. nodes | No. elements | No. partitions |
|-----------|-----------|--------------|----------------|
| mesh1 (reference) | 87 745 | 82 676 | 16 and 32 |
| mesh2 (half size) | 44 198 | 41 328 | 16 and 32 |
| mesh3 (double size) | 160 747 | 152 152 | 64 |

**Table 3.** Timings of different scalability test runs. All timings are given in seconds.

| Mesh (case) | Partitions | CPU time (s) | Wall-clock time (s) |
|-------------|------------|--------------|---------------------|
| mesh1 (single node) | 16 | 19 702 | 21 288 |
| mesh1 (reference) | 32 | 9016 | 9639 |
| mesh1 (half time-step size) | 32 | 14 319 | 16 351 |
| mesh2 (half size, 1 node) | 16 | 5035 | 6122 |
| mesh2 (half size, 2 nodes) | 32 | 3271 | 3683 |
| mesh3 (double size) | 64 | 14 817 | 15 800 |

the Intel MKL library. Message passing was enabled by linking to the Intel MPI library (version 5.1.0.097) provided on the system.

We want to emphasize that we only studied a limited set of problem sizes or computing resource configurations, and only single runs (no statistics) were performed. Results presented in the following thus have to be interpreted in view of the limitations. All runs performed are summarized in Table 3.

A comparison of a simulation performed with 16 cores (single compute node) with mesh2 (half size) and with 32 cores (two compute nodes) on mesh1 (reference) reveals a drop to 64 % of an ideal, linear weak scaling (increasing core numbers while maintaining the load/core) performance. This can be explained by adding additional latency to that part of the MPI communication that in the 32-core run has to be routed over the inter-nodal connection (Infiniband), whereas the 16-core run solely uses faster communication provided within a single compute node. Reassuringly, a similar value, namely 61 %, was obtained between runs on the double-size mesh (mesh3) with 64 cores on four compute nodes in relation to the reference problem (mesh1) run on 32 cores on two compute nodes. Studying the log files of the runs, it also becomes clear that the chosen GCR algorithm takes longer to converge with respect to the same convergence criteria if increasing the amount of mesh partitions. Another comparison with slightly less strict convergence criteria of the linear solution iteration algorithm led to a value of 84 %.

On the other hand, if looking at strong scalability (i.e., increasing core numbers while reducing load/core), doubling computational resources from 16 cores (single compute node) to 32 cores (inter-nodal) for the fixed-size smaller

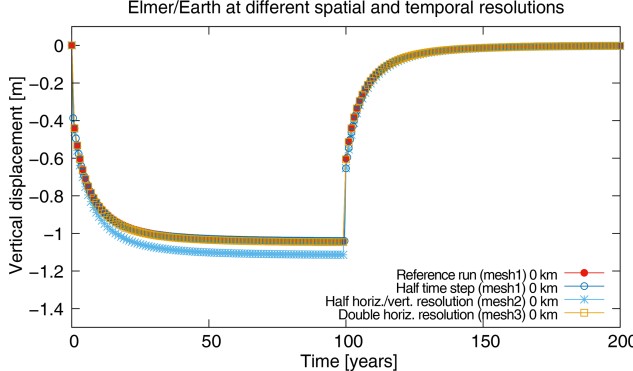

**Figure 6.** Vertical deformation at the centre (0 km) of Elmer/Earth simulations using different spatial and temporal resolutions.

problem (mesh2) revealed a speedup of 1.66, which is below the ideal value of 2 (half wall-clock time, by doubling of cores). For the larger reference problem (mesh1), we achieve a speedup of 2.2 if increasing from 16 (single node) to a 32 core utilizing two compute nodes of the reference run. We have not investigated the particular cause of this super-linear scaling further but can speculate on it: reducing the memory or core needed improves the possibility of fitting more data into the cache and thereby enabling faster memory access (i.e., avoiding cache misses) and hence – despite the added latency from inter-nodal communication – allowing for a general acceleration.

Despite applying the same solution method, it is not really possible to compare the performance of Elmer/Earth to ABAQUS, since the latter was run on a different platform using a regular mesh of 25 km TS4 constant horizontal mesh size. Computational performance was not the main motivation behind using ABAQUS for the benchmarking exercise; rather we wanted to use a model that could best replicate the geometry and equations used. Nevertheless, it is interesting to note that the run time of ABAQUS was in the range of 6 h using 32 cores on a high-end workstation, hence about twice the time of Elmer/Earth reference run on the same amount of cores of a larger Linux cluster. These run times should not be used in a direct comparison for computational performance, since ABAQUS was run on a mesh significantly larger (600 k nodes) than the one of Elmer/Earth. However, TABOO is using a completely different model approach, such that any comparison would be obsolete.

## 5.2 Accuracy with respect to mesh and time-step size

We further studied the accuracy and consistency of Elmer/Earth results with respect to spatial and temporal discretization sizes. To that end, we ran the same numerical setup on all three meshes given in Table 2.

Results are depicted in Fig. 6 and reveal that too low a spatial resolution (i.e., mesh2) – in that particular case in the horizontal as well as vertical direction – yields deforma-

tions that are too large. That might simply be because of too little resolution of the induced viscous deformation in under-resolved layers. The more finely resolved meshes (`mesh1` and `mesh3`) show very little deviations in results, thus indicating consistency of the model beyond a resolution of about 5 km mesh size at the centre of the geometry and the vertical structure depicted in Fig. 1b. On the other hand, increasing temporal accuracy by reducing the time-step size from 1 year to half a year did not reveal any significant difference in the result for similar setups to the reference run (`mesh1`).

## 6 Conclusions

We presented a newly implemented viscoelastic addition to the linear elasticity solver of the open-source finite-element package Elmer and its application to a flat-earth model. Robust projection of future ice-sheet change depends on coupled solid earth and ice dynamic processes at high spatial resolution, and Elmer/Earth provides a new open-source capability in conjunction with the existing ice-sheet model Elmer/Ice (Gagliardini et al., 2013). Elmer/Earth, on its own, provides a new tool for modelling viscoelastic solid earth deformation due to surface loading changes.

For the time being, Elmer/Earth is a so-called flat-earth model (Wu, 2004). In its current state it ignores sphericity and self-gravitational effects and neglects accounting for the deformation induced by the redistribution of ocean water masses. This introduces certain limitations on its applicability (Wu and Johnston, 1998). Consequently, future applications of this particular model version should be confined to regional studies of ice sheets or highly localized loads, such as glaciers and ice caps.

We benchmarked Elmer/Earth with another FEM code, ABAQUS, as well as a spherical viscoelastic normal mode code, TABOO, and these comparisons show good agreement in the range of deviation in solution method as well as numerical approaches.

Scaling figures presented in Sect. 5 are what one would expect from other parallel performance tests of Elmer. A good performance tuning strategy will have to make sure that a good ratio between partition size (i.e., computation mainly bounded by memory access) and communication between the different MPI tasks is obtained. OpenMP multi-threading is in principle available for certain modules in Elmer but is not implemented for the linear elasticity solver; however, it might be a potential way to boost performance within a single node (Byckling et al., 2017).

*Code availability.* Elmer (version 8.4) is available for download under GitHub (https://github.com/ElmerCSC/elmerfem, last access: 4 March 2020). The revision (SHA-1 14c19b6) used in this study can be retrieved from https://github.com/ElmerCSC/elmerfem/archive/14c19b681beb12df3a1d88fed9cd56a694b0cc92.zip (last access: 6 November 2019). TABOO is an open-source code available for download under GitHub (https://github.com/danielemelini/TABOO, last access: 4 March 2020). In this study we used version v1.1 (SHA-1 6163bec), which can be downloaded from https://github.com/danielemelini/TABOO/archive/v1.1.zip (last access: 6 November 2019). ABAQUS is proprietary software and needs a purchased license. We used the ABAQUS 2018 release in this study. Information on how to obtain the software can be found under https://www.3ds.com/products-services/simulia/products/abaqus/ (last access: 6 November 2019).

*Video supplement.* The animation (https://doi.org/10.5446/44086, Zwinger, 2019) shows the temporal deformation of the benchmark case for the reference run (on `mesh 1`) as discussed in the article. Deformations in the video shown are exaggerated by a factor of 10 000 CE3.

*Author contributions.* TZ helped developing and implementing the model setup and performing the computations in Elmer. GAN contributed to the design of the benchmark setup and performed the computations in ABAQUS and TABOO. JR implemented the altered model equations in the source code of Elmer. MAK conceived the study and consulted in the model implementation and contributed to the design of the benchmark test. All authors contributed to the paper.

*Competing interests.* The authors declare that they have no conflict of interest.

*Acknowledgements.* Development of the viscoelastic model was supported under the Australian Research Council's Special Research Initiative for Antarctic Gateway Partnership (Project ID SR140300001) and Discovery Project DP170100224. Part of the work of Thomas Zwinger was enabled by a visiting scientist scholarship from UTAS. This research was undertaken with the assistance and resources from the National Computational Infrastructure (NCI Australia), an NCRIS-enabled capability supported by the Australian Government. We want to express our gratitude to Peter Råback (CSC) for solving a problem with post-processing of Elmer/Earth data and Fredrik Robertsén (CSC) for the discussion on scalability test results. We are grateful to Giorgio Spada for making TABOO open source. We want to thank the two reviewers and the editor for constructive suggestions to improve the quality of this paper.

*Financial support.* This research has been supported by the Australian Research Council (grant nos. SR140300001 and DP170100224).

*Review statement.* This paper was edited by Thomas Poulet and reviewed by PingPing Huang and Surendra Adhikari.

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

**Remarks from the language copy-editor**

CE1    This is grammatically correct as it stands.

CE2    Placing the participle in front of the noun would be incorrect here.

CE3    Please note slight edits.

**Remarks from the typesetter**

TS1    Please note that we avoid the double use of parentheses within parentheses.

TS2    Please note that changes to numbers need to be approved by the editor. Please give an explanation of why this needs to be changed. We have to ask the handling editor for approval. Thanks.

TS3    Please give an explanation of why this needs to be changed. We have to ask the handling editor for approval. Thanks.

TS4    Please give an explanation of why this needs to be changed. We have to ask the handling editor for approval. Thanks.