# Peer review of "A new open-source viscoelastic solid Earth deformation module implemented in Elmer (v8.4)"

_Geoscientific Model Development, 2019_

## Referee Comment (RC1) · PingPing Huang (Referee) · 31 Dec 2019

The module proposed in this article is a good alternative in modeling Glacial Isostatic Adjustment (GIA) because it takes advantage of an open-source and free FEM package Elmer. The article is well written with a clear structure. I can support publishing the article if the author can provide more details of the method and more benchmark tests:

In section 2, what are the boundary conditions on internal boundaries and external surface for a flat Earth model and how are they implemented in the model ? In terms of solving Equation (9), what are the detailed form of the test and weighting functions

and what is the integration method ?

In section 3. when doing benchmark tests, it is more convincing that if the numerical solution can be compared with the analytical or semi-analytical solution. Therefore, it is good to compare the result from Elmer/Earth with that from normal-mode method for a Heaviside single harmonic load and a flat Earth model.

Below are some small issues:

Figure 1 and Figure 2: font size on axises is too small. Line 148: why does a high viscosity (e.g. 1 x1044 Pas) in Lithosphere enables an approximately elastic behaviour ? Why the viscosity of upper and lower mantle are set to be 1 x1018 Pas and 1 x1022 Pas respectively ?

––––––––––––––––––––––––––––––––

---

## Author Comment (AC1) · 20 Jan 2020

We thank PingPing Huang for this constructive review. Please, find our response inline to the suggestions:

**The module proposed in this article is a good alternative in modeling Glacial Isostatic Adjustment (GIA) because it takes advantage of an open-source and free FEM pack- age Elmer. The article is well written with a clear structure. I can support publishing the article if the author can provide more details of the method and more benchmark tests:**

We thank for the generally positive assessment of the reviewer and are grateful for the

work invested to improve the manuscript.

**In section 2, what are the boundary conditions on internal boundaries and external surface for a flat Earth model and how are they implemented in the model ?**
The main new aspect introduced in this paper is, that we are solving the complete set of equation over the whole domain and hence do not need to prescribe Winkler foundations. Accounting for layer discontinuities in material parameters is being taken care by the fact that the term – in contrary to the modification of commercial codes, where this only occurs in the body force – is appearing in the system matrix and produces the restoring force as a natural condition. We will expand the current explanation in the text by another sentence: *This means that we are able to impose discontinuities in parameters over elements anywhere in the discretized computing domain without placing Winkler foundation boundaries at layer interfaces. In other words, no boundary conditions have to be set at internal layer boundaries. By including this term in the weak formulation of the problem, the method then automatically applies the needed restoring force on element boundaries with jumps in material properties or gravity, without the need to place boundaries in the mesh.*
For the external boundaries we will add the following paragraph at the end of section 3.3: *We apply zero deformation at the side and the bottom boundaries. At the upper surface, we impose the load as described in the paragraph above.*

**In terms of solving Equation (9), what are the detailed form of the test and weighting functions and what is the integration method ?**
Indeed, this information was missing. We will add the following paragraph after equation (9):
*Equation (9) is solved using the standard Galerkin method with – in the case of the benchmark described in section 3 - first order basis functions. Apart from this particular choice, Elmer provides a variety of possible basis functions left to the choice of the user. The iteration for the viscous contribution is computed on the Gaussian integration points. In case of incompressibility, stabilization has to be applied by the residual free bubble method.*

**In section 3. when doing benchmark tests, it is more convincing that if the numerical solution can be compared with the analytical or semi-analytical solution. Therefore, it is good to compare the result from Elmer/Earth with that from normal-mode method for a Heaviside single harmonic load and a flat Earth model.**

We do not completely understand the request to test with spherical harmonics in combination with a flat-earth model and would need a detailed layout on a requested additional benchmark in terms of conditions imposed on Elmer/Earth, than currently given by the referee. It is our opinion that from the perspective of testing a flat-earth model, comparison to two other established models is sufficient for this manuscript.

**Below are some small issues:**

**Figure 1 and Figure 2: font size on axises is too small.**

We will provide figures with a larger font in a revised version of the manuscript.

**Line 148: why does a high viscosity (e.g. $1 \times 10^{44}$ Pas) in Lithosphere enables an approximately elastic behaviour?**

At the centennial timescale of our benchmark, the extreme value of the viscosity ensures that all loads are accommodated by an elastic response of the Lithosphere. Theoretically, this can be explained by an extreme resulting high value of the Maxwell-time (viscous relaxation time) of $10^{33}$ seconds ($10^{25}$ years). We will insert the following sentence:

*This can be justified by the Maxwell-time $t_m = \nu/\mu$ being of the order of $10^{33}$ seconds ($10^{25}$ years), which indicates that viscous effects in this layer only would be significant at timescales several order of magnitudes larger than the timing of the load signal in our experiment or even on timescales of glacial cycles on Earth.*

**Why the viscosity of upper and lower mantle are set to be $1 \times 10^{18}$ Pa s and $1 \times 10^{22}$ Pa s respectively ?**

As the latter value is a commonly used value for the mantle, the relatively low value of $1 \times 10^{18}$ Pa s for the upper mantle mainly is motivated to speed up the benchmark computation.

---

## Referee Comment (RC2) · PingPing Huang (Referee) · 22 Jan 2020

Referring to section 3, to be clearer: The results from Elmer/Earth for the loading presented in the paper are numerical solutions. Results from Abaqus and Taboo for the same loading are also numerical solutions. Is it possible to compare the numerical solutions from Elmer/Earth for a specific loading directly to the analytical solutions ? If so, the model presented in this paper would be more convincing.

---

## Referee Comment (RC3) · Surendra Adhikari (Referee) · 31 Jan 2020

This paper presents a new module implemented in Elmer, termed Elmer/Earth, that allows users to compute the solid Earth's deformational response to the applied surface loads. Given the observation of rapid response of solid Earth to ongoing ice mass loss and its possible stabilizing feedback to ice sheet dynamics (e.g., Barletta et al., 2018, doi: 10.1126/science.aao1447), Elmer/Earth is a welcome addition to Elmer particularly in light of Elmer/Ice (Gagliardini et al., 2013, doi: 10.5194/gmd-6-1299-2013) that can simulate evolving ice load subject to atmospheric and oceanic forcings.

For the reasons that follow, however, I am not so sure about the utility of this new mod-

ule to fulfill the purpose of improving our understanding of ice-sheet/solid-Earth interaction. Change in ice mass directly loads the underlying solid Earth, and hence, induces its deformation. Ice mass change also modulates the ocean mass, satisfying mass conservation in the Earth System. The change in ocean load contributes to the solid Earth deformation. Ignoring ocean load may underestimate the magnitude of modeled displacement field by about 10%, at least around the ice-bedrock-ocean interfaces. Elmer/Earth clearly lacks the ability to capture mass conserving ocean load induced by ice mass change, limiting its utility for the rigorous analysis of ice-sheet/solid-Earth interaction. Furthermore, both the ice and ocean mass change deform the geoid field, which further amplifies the strength of stabilizing feedbacks of the solid Earth to marine ice sheet dynamics. This element is also overlooked in the current version of Elmer/Earth. At a minimum the authors should acknowledge this limitation, with reference to recent works on the topic of ice-sheet/solid-Earth/sea-level interaction (e.g., Adhikari et al., 2020, doi: 10.5194/tc-2020-23). Elmer/Earth perhaps is more suitable for predicting local- or regional-scale hydrology (including ice) induced displacement fields.

I find that the lateral boundary conditions imposed in Elmer/Earth may be problematic for its application to continental-scale ice sheet. They have simply considered a "large enough" horizontal extent of the domain and set displacement vector to zero at the lateral boundaries. For Antarctic Ice Sheet, for example, one may require horizontal extent of the domain to be on the order of tens of thousands of kilometers. In such situations, the effects of Earth's sphericity are not certainly negligible unlike in the test case considered in the paper (line 135). Either a justification about this inconsistency or an acknowledgement of this limitation is required.

Providing a bit more elaborative description of Theory (Section 2) would be useful, especially for those who are not familiar with Wu (2004, doi: 10.1111/j.1365-246X.2004.02338.x). Section 2 of the Wu paper is very informative, and all I see in this paper is a list of equations (with minimal explanation) that are deduced from the

Wu paper for the case of incompressible viscoelastic Earth that lacks self-gravitation and sphericity. Also, missing in this section is the (mathematical) description of boundary conditions.

A few suggestions on the usage of terminologies: Visco-elastic => viscoelastic; Earth => solid Earth; Finite Element => finite element. So, the title would be "...viscoelastic solid Earth module..."

There is something about the word "flat-earth" that does not look right to me (especially in the era of social media). I would consider avoiding it.

Line 25: Is this timescale required for a "complete" relaxation of mantle? Or e-folding relaxation?

Around line 30: Evolving bedrock also modulates the gravitational driving stress of the ice sheet and hence its dynamics (see Figure 6 of Adhikari et al., 2014, doi: 10.5194/se-5-569-2014)

Line 33: Provide references, e.g. Schoof (2007, doi: 10.1029/2006JF000664).

Line 41: Best performing => in terms of computational ability? Or, its ability to match, say, bedrock GPS data?

Line 41: widely used => Not sure about this(!). Provide references at least. Again, for the reason I noted in the beginning of this review, even if this method is "widely" used it certainty is not the most accurate one.

Line 53: suggested rewording: "...a full Stokes ice sheet model capable of yielding high wave number loads that is essential to model high-res solid Earth rebound"

Line 56: high gradients => of what?

Line 65: Cauchy stress => Cauchy stress tensor

Line 66 (and elsewhere): deformation vector => displacement vector

[Figure]

Line 69: Suggested rewording: "...motion that conserves linear momentum for a non-gravitating... layered material..."

Line 74: Why do you need to introduce d_z? Simply write "...vector product e_z . d"

Line 79: Given the limited description of Theory (as noted above), not sure I follow what you exactly mean by "avoids singularity... as Poisson ratio approaches $\frac{1}{2}$". Either refer this statement to some equation or delete it altogether.

Line 84: Why 10 layers? Be generic.

Line 121: The model is fixed in all directions => What do you mean by fixed? Dirichlet conditions with zero displacement? Again, you need to talk about boundary conditions in Theory section.

Line 135: "sea-level equation" => Need to provide a qualitative description of what it means if it is relevant at all (else simply delete the sentence). Not all readers would get what it means.

Line 137/138: Looks like it is 40,000 km (see Figure 1); and that would be 800 times larger? Again, Is the sphericity effect negligible for such a large spatial scale?

Line 144: "has a resolution equivalent" => "has a spectral resolution equivalent"

Figures 1/2: Combine these as Figure 1a and 1b?

Line 152: zero time => t = 0

New Figure (that corresponds to Figure 4): Would be useful to show a new figure with displacement vs. distance away from the load center for select times (including at t=0 to show the elastic displacement fields).

Around line 175: Acknowledge that ABAQUS uses compressible Earth (see Table 1 caption). Elmer/Earth solves for incompressible Earth.

Section 5.2: I was wondering whether you maintain the same mass for different mesh

[Figure]

experiments (by adjusting ice height). Otherwise the discrepancy in solutions may be (at least partly) due to the fact that you are loading the solid Earth with slightly different loads (i.e., net mass) and not necessarily do to the coarseness or fineness of computational mesh.

Figure 6 caption: Vertical deformation => Vertical displacement

Lines 241: For the reasons noted early on, I am afraid that the utility of Elmer/Earth to accurately capture solid Earth's feedback to ice sheet dynamics (within Elmer/Ice) is limited. At least, it should be acknowledged. I would highlight the utility of Elmer/Earth for general (regional/local) loading studies (hydrology, ice load, atmosphere loads, etc).

Last paragraph: I am not sure whether this should be part of the conclusion. It may be sufficient to say that Elmer/Earth performs well in parallel computation.

---

## Author Comment (AC2) · 7 Feb 2020

**Response to reviewer 2 (Surendra Adhikari)**

**This paper presents a new module implemented in Elmer, termed Elmer/Earth, that allows users to compute the solid Earth's deformational response to the applied surface loads. Given the observation of rapid response of solid Earth to ongoing ice mass loss and its possible stabilizing feedback to ice sheet dynamics (e.g., Barletta et al.,2018, doi: 10.1126/science.aao1447), Elmer/Earth is a welcome addition to Elmer particularly in light of Elmer/Ice (Gagliardini et al., 2013, doi: 10.5194/gmd-6-1299-2013) that can simulate evolving ice load subject to atmospheric and oceanic forcings.**

We thank for the generally positive assessment of the reviewer and are grateful for the detailed work invested to improve the manuscript. Please, find our response inline to the suggestions:

**For the reasons that follow, however, I am not so sure about the utility of this new module to fulfill the purpose of improving our understanding of ice-sheet/solid-Earth interaction. Change in ice mass directly loads the underlying solid Earth, and hence, induces its deformation. Ice mass change also modulates the ocean mass, satisfying mass conservation in the Earth System. The change in ocean load contributes to the solid Earth deformation. Ignoring ocean load may underestimate the magnitude of modelled displacement field by about 10%, at least around the ice-bedrock-ocean interfaces. Elmer/Earth clearly lacks the ability to capture mass conserving ocean load induced by ice mass change, limiting its utility for the rigorous analysis of ice-sheet/solid-Earth interaction. Furthermore, both the ice and ocean mass change deform the geoid field, which further amplifies the strength of stabilizing feedbacks of the solid Earth to marine ice sheet dynamics. This element is also overlooked in the current version of Elmer/Earth. At a minimum the authors should acknowledge this limitation, with reference to recent works on the topic of ice-sheet/solid-Earth/sea-level interaction (e.g., Adhikari et al., 2020, doi: 10.5194/tc-2020-23). Elmer/Earth perhaps is more suitable for predicting local- or regional-scale hydrology (including ice) induced displacement fields.**

That is a correct statement. We have clarified that we recommend the model's application only in a regional context by adding the following sentence in the conclusions:

> *Elmer/Earth for the time being is a so-called flat-earth model (Wu, 2004). In its current state it ignores sphericity and self-gravitational effects as well as neglects to account for the deformation induced by redistribution of ocean water masses. Consequently, future applications of this particular model version should be confined to regional studies of ice-sheets or highly localized loads, such as glaciers and ice-caps.*

For the suggested reference (Adhikari et al., 2020), we would like to point out that this is a paper that currently is in review and was not available for citation at the time this manuscript was submitted.

**I find that the lateral boundary conditions imposed in Elmer/Earth may be problematic for its application to continental-scale ice sheet. They have simply considered a "large enough" horizontal extent of the domain and set displacement vector to zero at the lateral boundaries. For Antarctic Ice Sheet, for example, one may require horizontal extent of the domain to be on the order of tens of thousands of kilometers. In such situations, the effects of Earth's sphericity are not certainly negligible unlike in the testcase considered in the paper (line 135). Either a justification about this inconsistency or an acknowledgement of this limitation is required.**

We will mention the neglected sphericity and its limitations in the text presented in the conclusions (see above) and also drop a note on this limitation in the new text describing the implementation of boundary conditions (see further below and response to comment on line 137/138).

**Providing a bit more elaborative description of Theory (Section 2) would be useful, especially for those who are not familiar with Wu (2004, doi: 10.1111/j.1365-246X.2004.02338.x). Section 2 of the Wu paper is very informative, and all I see in this paper is a list of equations (with minimal explanation) that are deduced from Wu paper for the case of incompressible viscoelastic Earth that lacks self-gravitation and sphericity. Also, missing in this section is the (mathematical) description of boundary conditions.**
We already described the complete model as derived in Section 3 of Wu (2004). In order to make it easier for the reader to evaluate which simplification have been taken place, we will add the following paragraph to the existing presentation of (currently) equation (4) in Section 2:

> *The linearized elastic equation of motion for earth deformation (Wu, 2004) is given by*
> $$\nabla \cdot \boldsymbol{\tau} - \nabla(\rho_0 \boldsymbol{g}_0 \cdot \mathbf{d}) - \rho_1 \boldsymbol{g}_0 - \rho_0 \nabla \phi_1 = \mathbf{0} \quad (4)$$
> *Where $\rho_0$ and $\boldsymbol{g}_0$ are hydrostatic background density and gravity, respectively, and $\rho_1$ is the perturbed density. The direction of $\boldsymbol{g}_0$ is in negative radial direction. According to Wu (2004, section 3), a flat-earth model is derived from (4) by assuming incompressibility and ignoring self-gravitational effects (i.e., redistribution of mass), making the third and fourth term vanish. Further, sphericity is ignored, leading to changes aligned with the unit vector of a Cartesian system in vertical direction, $\boldsymbol{e}_z$. This leads to the equation of motion for a non-self-gravitating flat-earth model with layer-wise constant material. It reduces to a balance between the divergence of the stress (first term) and a restoring force due to the advection of pre-stress of the material (Wu, 2004)*
> $$\nabla \cdot \boldsymbol{\tau} - \rho g \nabla(\boldsymbol{e}_z \cdot \boldsymbol{d}) = \mathbf{0} \quad (5)$$
> *Here, $\rho = \rho_0$ and $g = |\boldsymbol{g}_0|$ is the magnitude of the local acceleration by gravity. …*

For the boundary condition, we will add the following sentence directly after (currently) equation (9) in Section 2.1:

> *The system is completed by boundary conditions that are either provided by a value for any component of the stress-vector, $\mathbf{t} = \boldsymbol{\tau} \cdot \boldsymbol{n}$, in the second integral (Neumann condition) of equation (10 , former 9) or by imposing a value for any component of the deformation vector, $\mathbf{d}$ (Dirichlet condition).*

And we explicitly state the boundary conditions for the benchmark in Section 3:

> *At the free surface we apply the load of the disc for the first 100 years into the simulation. Thereafter, the natural boundary condition, namely a vanishing stress vector, applies to the whole surface. At all other boundaries we impose a zero-deformation condition, i.e., $\mathbf{d} = \mathbf{0}$.*

**A few suggestions on the usage of terminologies: Visco-elastic => viscoelastic; Earth=> solid Earth; Finite Element => finite element. So, the title would be "...viscoelastic solid Earth module..."**
We will change accordingly

**There is something about the word "flat-earth" that does not look right to me (especially in the era of social media). I would consider avoiding it.**
We are well aware of the distorted connotation of the term in some communities with a – let us say - weird alternative approach to Earth science. Yet, the term has a scientific significance as it is used in the main reference (Wu, 2004) and it would be difficult for the reader to make the connection therein. We have retained the standard use in model studies of 'flat earth'. We are sure that the readers of this journal will apply the correct interpretation of this word.

**Line 25: Is this timescale required for a "complete" relaxation of mantle? Or e-folding relaxation?**
This is referring to a typical "Maxwell" relaxation time, rather than a complete relaxation. It is intended to be merely an indication of the timescales one could expect to observe viscoelastic relaxation over due to changes in ice loading with the point that some regions experience a very quick response due to low viscosity underlying mantle. We do not feel it is necessary to include this in the text as it may be confusing for some readers. We have however quantified the more rapid timescale as follows:

> *…although typically thought to occur over several thousands of years (Whitehouse, 2018, and references therein), recent studies have shown some regions undergoing much more rapid (decadal) rebound in response to present-day changes (Nield et al., 2014; Barletta et al., 2018)….*

**Around line 30: Evolving bedrock also modulates the gravitational driving stress of the ice sheet and hence its dynamics (see Figure 6 of Adhikari et al., 2014, doi:10.5194/se-5-569-2014)**
We include this reference

**Line 33: Provide references, e.g. Schoof (2007, doi: 10.1029/2006JF000664).**
We will include this citation on line 33.

**Line 41: Best performing => in terms of computational ability? Or, its ability to match, say, bedrock GPS data?**
Both, but mainly in terms of numerical performance. The shortcomings on physics are mentioned in the sentence to follow. We will add:

> *Of these, Le Meur and Huybrechts (1996) found the best performing is the "ELRA" model (elastic lithosphere with relaxing asthenosphere) which is widely used in ice-sheet modelling, mainly due to its simplicity and fast computations.*

**Line 41: widely used => Not sure about this(!). Provide references at least. Again, for the reason I noted in the beginning of this review, even if this method is "widely" used it certainty is not the most accurate one.**
It is in our view an up until recently widely used method. We will add another reference (Greve, 2001) to the already existing references (Le Meur and Huybrechts, 1996; Rutt et al., 2009). In no way do we claim that its popularity compensates for its shortcomings, which we believe are sufficiently mentioned in the sentence that follows: *However, Bueler et al. (2007) found significant differences …*

**Line 53: suggested rewording: "...a full Stokes ice sheet model capable of yielding high wave number loads that is essential to model high-res solid Earth rebound"**
We reword accordingly

**Line 56: high gradients => of what?**
We reformulate:

> *Finite Elements have the advantage that they in general can use unstructured meshes in order to provide the needed resolution in regions where either physics or geometry demand it while keeping the model size limited.*

**Line 65: Cauchy stress => Cauchy stress tensor**
We will change the text according to this suggestion

**Line 66 (and elsewhere): deformation vector => displacement**
We will change the text according to this suggestion

**Line 69: Suggested rewording: "...motion that conserves linear momentum for a non-gravitating...layered material..."**
As shown above, we completely dropped the brackets.

**Line 74: Why do you need to introduce d_z? Simply write "...vector product e_z . d"**
We dropped this sentence.

**Line 79: Given the limited description of Theory (as noted above), not sure I follow what you exactly mean by "avoids singularity...as Poisson ratio approaches". Either refer this statement to some equation or delete it altogether**
We reformulate to:

> … avoids the singularity of the first Lamé parameter $\lambda = E\eta/((1+\eta)(1-2\eta))$ in the limit of the Poisson ratio approaching $\eta \rightarrow \frac{1}{2}$ (e.g., Greve and Blatter, 2009; p 41).

**Line 84: Why 10 layers? Be generic.**
Added:

> … of several layers due to the resolution of changing material parameters.

**Line 121: The model is fixed in all directions => What do you mean by fixed? Dirichlet conditions with zero displacement? Again, you need to talk about boundary conditions in Theory section.**
We reformulate:

> …and has zero deformation imposed on its lateral boundaries.

**Line 135: "sea-level equation" => Need to provide a qualitative description of what it means if it is relevant at all (else simply delete the sentence). Not all readers would get what it means**
We delete this sentence.

**Line 137/138: Looks like it is 40,000 km (see Figure 1); and that would be 800 times larger? Again, Is the sphericity effect negligible for such a large spatial scale?**
No, $2 \times 10^6$ m make 2,000 km in each direction, which gives 4,000 km and not 40,000 km in total span. Following the request of reviewer 1, we will increase the resolution of the annotation in Fig. 1 in order to make it better readable.
The flat-earth approach, with the necessary large lateral extents, has been shown to be accurate when computing deformation for ice loads as large as the Laurentide ice sheet (Wu and Johnston, 1998). Nevertheless, we are computing a benchmark here and – as should be implicitly clear by load's small size of only 50 km in radius - in no way claim that this resembles a continental ice-sheet. Placing the boundaries so far out simply avoids any influence of boundaries on the displacement close to the centre. That motivation is reflected by the already existing line (and reference therein): *This distance is 80 times the diameter of the test load which is more than sufficient to allow mantle deformation below the ice load (Steffen et al., 2006).*

**Line 144: "has a resolution equivalent" => "has a spectral resolution equivalent"**
We will change the text according to this suggestion

**Figures 1/2: Combine these as Figure 1a and 1b?**
We will evaluate whether it is possible to combine them and still introduce larger annotation (as requested by reviewer 1).

**Line 152: zero time => t = 0**
We will change the text according to this suggestion

**New Figure (that corresponds to Figure 4): Would be useful to show a new figure with displacement vs. distance away from the load center for select times (including at t=0 to show the elastic displacement fields).**

We will include such a figure.

**Around line 175: Acknowledge that ABAQUS uses compressible Earth (see Table 1 caption). Elmer/Earth solves for incompressible Earth.**

This was an error in the text. Whilst ABAQUS can implement some aspects of compressibility (e.g. with a Poisson's ration < 0.5) it cannot change the density of elements with time. We use a high Poisson ratio in the benchmarking case to simulate incompressibility. We therefore remove "compressible" from Table 1 caption.

**Section 5.2: I was wondering whether you maintain the same mass for different experiments (by adjusting ice height). Otherwise the discrepancy in solutions maybe (at least partly) due to the fact that you are loading the solid Earth with slightly different loads (i.e., net mass) and not necessarily do to the coarseness or fineness of computational mesh.**

We are not completely sure we understand this question. We assume that it is about whether we change the net mass of the ice-disc in order to compensate for a lower resolution in the centre of the domain. We do not think that this is the case, since we impose a load and the mesh resolution (i.e. what is being tested) is inherent in how the disc load is represented in the model.

**Figure 6 caption: Vertical deformation => Vertical displacement**

We will change the text according to this suggestion

**Lines 241: For the reasons noted early on, I am afraid that the utility of Elmer/Earth to accurately capture solid Earth's feedback to ice sheet dynamics (within Elmer/Ice) is limited. At least, it should be acknowledged. I would highlight the utility of Elmer/Earth for general (regional/local) loading studies (hydrology, ice load, atmosphere loads, etc).**

We included a statement on this in the conclusions (see earlier).

**Last paragraph: I am not sure whether this should be part of the conclusion. It may be sufficient to say that Elmer/Earth performs well in parallel computation.**

As this is a technical paper that also should inform the reader on the applicability – including computational aspects – of the new code, we believe that this paragraph has a place in the conclusions.

**New References**

R. Greve, Glacial Isostasy: *Models for the Response of the Earth to Varying Ice Loads*, in B. Straughan, R. Greve, H. Ehrentraut, and Y. Wang (edt.), Continuum Mechanics and Applications in Geophysics and the Environment, Springer, Berlin, Germany etc., 393 pp. (2001). ISBN: 3-540-41660-9

R. Greve and H. Blatter, *Dynamics of Ice Sheets and Glaciers*, Springer, Berlin, Germany (2009) DOI 10.1007/978-3-642-03415-2

Wu, P. & Johnston, P., 1998. *Validity of Using Flat-Earth Finite Element Models in the Study of Postglacial Rebound*. in Dynamics of the Ice Age Earth, pp. 191-202, ed. Wu, P. Trans Tech Publications Ltd, Switzerland.

Schoof, C., 2007*. Ice sheet grounding line dynamics: Steady states, stability, and hysteresis*, Journal of Geophysical Research: Earth Surface, 112, F03S28.